# A Critical Analysis on the Sensitivity Enhancement of Surface Plasmon Resonance Sensors with Graphene

**DOI:** 10.3390/nano12152562

**Published:** 2022-07-26

**Authors:** Aline dos Santos Almeida, Dario A. Bahamon, Nuno M. R. Peres, Christiano J. S. de Matos

**Affiliations:** 1School of Engineering, Mackenzie Presbyterian University, São Paulo 01302-907, Brazil; alinesaalm@gmail.com (A.d.S.A.); dario.bahamon@mackenzie.br (D.A.B.); 2MackGraphe-Graphene and Nanomaterials Research Institute, Mackenzie Presbyterian Institute, São Paulo 01302-907, Brazil; 3Physics Department, Minho University, Campus of Gualtar, 4710-057 Braga, Portugal; peres@fisica.uminho.pt

**Keywords:** surface plasmon resonance, sensitivity, graphene, biosensors

## Abstract

The use of graphene in surface plasmon resonance sensors, covering a metallic (plasmonic) film, has a number of demonstrated advantages, such as protecting the film against corrosion/oxidation and facilitating the introduction of functional groups for selective sensing. Recently, a number of works have claimed that few-layer graphene can also increase the sensitivity of the sensor. However, graphene was treated as an isotropic thin film, with an out-of-plane refractive index that is identical to the in-plane index. Here, we critically examine the role of single and few layers of graphene in the sensitivity enhancement of surface plasmon resonance sensors. Graphene is introduced over the metallic film via three different descriptions: as an atomic-thick two-dimensional sheet, as a thin effective isotropic material (same conductivity in the three coordinate directions), and as an non-isotropic layer (different conductivity in the perpendicular direction to the two-dimensional plane). We find that only the isotropic layer model, which is known to be incorrect for the optical modeling of graphene, provides sizable sensitivity increases, while the other, more accurate, models lead to a negligible contribution to the sensitivity.

## 1. Introduction

Lately, much attention has been paid to the development of new chemical and biological sensors based on low cost and fast diagnosis [1,2,3,4]. Surface plasmon resonance (SPR) is a high sensitivity, real time, and label-free technique with great potential for sensing applications [5,6]. Surface plasmon polaritons (SPPs) are surface waves propagating at a metal/dielectric interface and arise from the electromagnetic field coupling to dipolar excitation of free electrons in the metal. In the resonance condition, these waves are highly sensitive to changes in the dielectric function of the surrounding environment [6,7]. In particular, for SPR biosensors, reversible adsorption of biomolecules (analyte) onto the sensing surface (metal/dielectric interface) changes the local refractive index (RI), leading to changes in the SPR condition [8].

The primary design feature of a SPR sensor is the sensitivity, which, depending on the interrogation method, is defined as the ratio between the shift of the SPR resonance angle or wavelength and the change in the refractive index of the analyte. An increased sensitivity translates into the possibility of the detection of analytes with low molecular weight and low concentrations [9]. In this regard, it has been predicted (always via numerical methods) that integrating graphene to conventional plasmonic materials in the visible would enhance the sensitivity of the SPR sensors. Specifically, it is believed that the sensitivity increases with the number of layers [5,9,10,11]. For example, Wu and coworkers [10] reported that a gold SPR sensor in the Kretschmann configuration with L layers of graphene would be (1+0.025L)γ (where γ>1) times more sensitive than a traditional gold SPR sensor. Along the same lines, for a fiber Au-based SPP sensor, Fu et al. reported that 20 layers of graphene would improve the device sensitivity up to 50% [12]. These predictions ignited systematic studies of SPP sensors with graphene on top of other metals [13,14,15], graphene–metal hybrid structures [16,17], as well as the use of other 2D materials [18,19,20,21]. In all these theoretical works, enhanced sensitivity was predicted.

In all of the previous reports, the sensitivity of the N layer SPR sensor was calculated through the transfer matrix method, where each layer of graphene is modeled by an effective thickness. Note that this means that after the multiplication of the matrices of the individual layers, we end up with a thicker homogeneous sample, in opposition to the experimental evidence that showed that, in few-layer graphene, each monolayer is minimally perturbed by the presence of the other layers [22,23]. With the above in mind, in this work, to access the actual impact of graphene on the SPR sensor’s sensitivity, we modeled graphene in a number of ways and compared the obtained results. First, graphene is modeled as a surface conductivity in which case it is treated as a strictly 2D material. Subsequently, graphene is treated as an ultrathin isotropic film, and, finally, as an ultrathin anisotropic film. Here, our analysis is focused only in the graphene’s out-of-plane anisotropy, unlike works that have already studied graphene in plane anisotropy, such as [24], where changes in the graphene’s transparency in the visible range are assigned to its strain-induced optical anisotropy. Our results show that the selected modeling approach directly impacts on the observed sensitivity enhancement, with significant enhancements obtained only in the isotropic case. Furthermore, in the case of an isotropic ultrathin layer, the results highlight the importance of the choice of the value for graphene’s refractive index, with results diverging more than ∼50% among them.

## 2. Model Description

To avoid complications with particular design parameters of specific SPR sensors, but without loss of generality, the structure of the studied sensor is based on the Kretschmann configuration. As sketched in Figure 1, it consists of a multilayer structure: a SF11 glass prism, a 50 nm-thick gold film, a graphene layer, and the analyte. In this configuration, the SPR resonance appears as a dip in the reflectance of *p*-polarized light impinging on the prism at a certain angle larger than the critical angle of the prism–analyte interface. Thus, the SPR condition is easily calculated through the reflectance of the multilayer structure ($R=|r_{1\ldots N}{|}^{2}$). Note that we treat the structure as a three-layer system when graphene is modeled as a surface conductivity, and as a four-layer system when graphene is taken to be a thin film.

For the three-layer system (2D graphene model; see Figure 2a), the first layer is the SF11 glass prism (semi-infinite) with refractive index $n_{1}$ = 1.7786 coated by a gold film with refractive index $n_{2}$ = 0.1834 + 3.4332*i* [25], and the third layer is the aqueous analyte that is considered semi-infinite (water refractive index taken to be 1.332). In the four-layer case (3D graphene model presented in Figure 2b), graphene is considered the third layer with thickness $d_{g}$ and refractive index $n_{3}^{g}$, while the semi-infinite region of the analyte is now considered the fourth layer, with refractive index $n_{4}$. The $n_{3}^{g}$ parameter is discussed in detail below, for the isotropic and anisotropic cases. The reflection coefficients for the three- and four-layer systems are respectively given by
(1)$$\begin{array}{c} r_{123}=\;\frac{r_{12\;}\;+\;r_{23}\;e^{i2\phi_{2}}\;}{{1\;+\;r}_{12}\;r_{23}\;\;e^{i2\phi_{2}}} \end{array}$$
and
(2)$$\begin{array}{c} r_{1234}=\frac{\left(r_{12}+r_{23}e^{i2\phi_{2}}\right)\;+\left(r_{12}r_{23}+e^{i2\phi_{2}}\right)r_{34}e^{i2\phi_{3}}}{\left({1+r}_{12}\;r_{23}\;e^{i2\phi_{2}}\right)+\left(r_{23}+r_{12\;}e^{i2\phi_{2}}\right)r_{34}e^{i2\phi_{3}}}, \end{array}$$
where $r_{ij}$ are the Fresnel reflection coefficients for *p*-polarized waves between layers *i* and *j*, $\phi_{i}=k_{zi}d_{i}$ and $k_{zi}=\sqrt{n_{i}^{2}\frac{\omega^{2}}{c^{2}}-k_{xi}^{2}}$. Given that the layered systems present translational symmetry along the *x*-direction, $k_{xi}=k_{x}=n_{1}\left(\frac{\omega }{c}\right)\sin\theta_{i}$. ω and *c* are, respectively, the angular frequency and the speed of light in vacuum.

The angle corresponding to the reflectance minimum is called SPR angle $\left(\theta_{SPR}\right)$. The sensing mechanism consists of monitoring the $\theta_{SPR}$ shift as the refractive index of the analyte (sensing medium) changes [26]. We assume that the refractive index of the aqueous analyte $n_{3(4)}$ varies from 1.332 to 1.37 in steps of $\Delta n=5\times 10^{-6}$. The sensor’s sensitivity is defined as $S=\frac{d\theta_{SPR}}{dn_{3(4)}}$, in units of ${}^{\circ }$/RIU (where RIU stands for refractive index units), and is calculated for each studied case. To clearly access the effect of the graphene modeling on the performance of the sensitivity, the thickness of the gold layer ($d_{2}=50$ nm) and the wavelength of the incident light (λ=633 nm) are kept constant.

### Graphene Modeling

In the three-layer case, graphene is modeled as an atomically thin film. Its surface conductivity ($\sigma_{g}$) is introduced in the boundary conditions of the electromagnetic fields at the metal–analyte interface to obtain the Fresnel reflection coefficient [27]
(3)$$\begin{array}{c} r_{23}=\frac{k_{z_{2}}\epsilon_{3}\;-\;k_{z_{3}}\epsilon_{2}\;+\;\frac{k_{z_{2}}k_{z_{3}}\sigma_{g}}{\epsilon_{0}\omega }}{k_{z_{2}}\epsilon_{3}\;+\;k_{z_{3}}\epsilon_{2}\;+\;\frac{k_{z_{2}}k_{z_{3}}\sigma_{g}}{\epsilon_{0}\omega }}. \end{array}$$

In this expression, $\epsilon_{0}$ is the vacuum permittivity. In the visible region of the electromagnetic spectrum, ${\sigma_{g}\approx \sigma }_{0}\;=\;\frac{e^{2}}{4\hbar }$ [22,27], with *e* being the electron charge and *ℏ* the reduced Planck constant.

The refractive index of graphene in the 3D isotropic model is obtained from the surface conductivity ($\sigma_{g}$). To do this, we have to bear in mind that the optical response of a thin film can be described by one of the four volumetric complex response functions: the optical conductivity ($\sigma^{3D}$), the optical susceptibility ($\chi^{3D}$), the dielectric function (ε) or the refractive index (*n*). The relation among all of them is well established: $\chi^{3D}=i\sigma^{3D}/\varepsilon_{0}\omega$, $\varepsilon =1+\chi^{3D}$ and $n=\sqrt{\varepsilon }$ [28]. The surface conductivity, in the 2D model, can be taken to be the product of the corresponding volumetric response function and the effective layer thickness ($\sigma_{g}=\sigma^{3D}d_{g}$) [29]. Thus, by taking the sheet conductivity of graphene to be a known parameter, we can write the refractive index of 3D graphene as
(4)$$\begin{array}{c} n_{3}^{g}=\sqrt{1+i\frac{\sigma_{g}}{\epsilon_{0}\omega d_{g}}}=2+1.7119i. \end{array}$$

Graphene’s effective thickness can be assumed to be the interlayer distance in graphite: $d_{g}=0.34$ nm [10,30,31,32].

From the experimental perspective, the optical constants of graphene were obtained after fitting processes of reflectance, transmittance and ellipsometry measurements [33,34,35,36]. However, the agreement between different experimental approaches is not so good. This can partially be explained by the fact that experimental data were fitted to distinct mathematical models, allowing/restraining thickness dependence. Thus, to avoid any further complications, in this work, we decided to work with two values of the refractive index of graphene: the one obtained from a fully theoretical approach, from Equation (Equation 4) ($n_{3}^{g}=2+1.7119i$), and the one used by Wu and coworkers ($n_{3}^{g}=3+1.1491i$) that can be traced back to the experimental work of M. Bruna [35].

In the anisotropic case, graphene is taken to have different in-plane and out-of-plane values for the diagonal dielectric tensor components. Specifically, for the in-plane response, the refractive index was considered to have the same value as in the isotropic case $n_{3\mathrm{in}}^{g}=n_{xx}=n_{yy}=\sqrt{\epsilon_{3\mathrm{in}}^{g}}=\sqrt{1+i\frac{\sigma_{g}}{\epsilon_{0}\omega d_{g}}}$. The out-of-plane refractive index was assumed to be $n_{3z}^{g}=\sqrt{\epsilon_{3z}^{g}}=\sqrt{2.5}$ [31,37,38]. Considering that the reflecting surface of the anisotropic thin film coincides with graphene’s basal plane, the coupling between the *s*- and *p*-polarized waves is zero. This ensures that the reflectance of the four-layer system with the anisotropic graphene continues to be calculated by Equation (Equation 2), but with $\phi_{3}=k_{z3e}d_{3}$,
(5)$$\begin{array}{c} k_{z3e}=\sqrt{\left(\epsilon_{3\mathrm{in}}^{g}\frac{\omega^{2}}{c^{2}}-\frac{\epsilon_{3\mathrm{in}}^{g}}{\epsilon_{3z}^{g}}k_{x}^{2}\right)}, \end{array}$$
and
(6)$$\begin{array}{c} r_{i3}=\frac{{\left(n_{3\mathrm{in}}^{g}\right)}^{2}k_{zi}-n_{i}^{2}k_{z3e}}{{\left(n_{3\mathrm{in}}^{g}\right)}^{2}k_{zi}+n_{i}^{2}k_{z3e}}. \end{array}$$

For clarity, we now briefly summarize the main structural and material aspects of the SPR sensor modeling. We model the sensor as a layered structure. When graphene is modeled as a fully 2D material, the SPR sensor is composed of three layers (prism/gold/analyte), the contribution of graphene is included in the Fresnel reflection coefficient of the gold/analyte interface (Equation (Equation 3)) through graphene’s surface conductivity, which, at the studied frequency, is taken to be $\sigma_{g}=\sigma_{0}=e^{2}/4\hbar$. When graphene is modeled as a 3D material, the SPR sensor has four layers (prism/gold/graphene/analyte). In this case, there are two graphene parameters: (a) the effective thickness, which can be safely considered to be the interlayer distance $d_{g}=0.34$ nm, and (b) the refractive index of graphene $n_{3}^{g}$. For the isotropic case, we consider only two values for the refractive index; $n_{3}^{g}=2+1.7119i$ and $n_{3}^{g}=3+1.1491i$. In the anisotropic case, we keep the same index values for the in-plane response, and consider the out-of-plane index to be $n_{3z}^{g}=\sqrt{2.5}$.

## 3. Results and Discussion

The effect of the different approaches used to model the graphene monolayer is presented in Figure 3a, which shows the reflectance for the three-layer ($R_{123}$) and the four-layer ($R_{1234}$) systems as functions of the angle of incidence, $\theta_{i}$, for two different analyte refractive indices: $n_{3(4)}=1.332\;\mathrm{and}\;1.342$ (red and black curves, respectively). The continuous lines correspond to the response of the sensor without graphene; the dashed lines present the response with graphene as a sheet conductivity (2D model); the dotted lines stand for the 3D isotropic case; and the dashed-dotted lines represent the 3D anisotropic model.

At first sight, the four models produce similar reflectance curves, with graphene only slightly modifying the response of the sensor. However, the zoom in the reflectance curves provided in Figure 3b shows observable differences produced by the different graphene models, particularly in the shift of $\theta_{SPR}$ with the analyte refractive index, as well as in the amplitude of the reflectance at the plasmon resonance condition. For example, in the system based only on gold’s SPR, the SPR corresponds to a minimum reflectance of ∼0.71% for the analyte refractive index of 1.332 (solid red curve in Figure 3). By adding the graphene monolayer and modeling it either through the 2D model or the anisotropic thin film model, the reflectance minimum increases to ∼1.9% due to the absorption by the sheet of graphene, while modeling graphene as a thin isotropic layer produces not only the largest shift of $\theta_{SPR}$, but also the largest change in the reflectance minimum, which increases to ∼2.6%.

However, the most important result of the present work is to quantify the sensitivity of the sensor (i.e., the SPR shift with analyte index change), which is shown in Table 1 for the different graphene models ($S_{gr}$), as well as for the sensor without graphene ($S_{Au}$). The sensitivity increase provided by graphene, given by $\Delta S=\left(\left(S_{gr}-S_{Au}\right)/S_{Au}\right)\times 100$, is also presented for each case. When modeling monolayer graphene as an atomic sheet (2D model) or as an anisotropic thin film (3D anisotropic) there is minimal change in the system’s sensitivity relative to case of gold only. In stark contrast, modeling monolayer graphene as an isotropic film, which is known to lead to unrealistic results, as discussed [31], yields a sensitivity increase that is 3 orders of magnitude higher than the 2D model.

Having established that the way a single sheet of graphene is modeled affects the obtained (theoretical) sensor sensitivity, we now investigate the sensitivity as a function of the number of graphene layers. To model more than one layer, previous works treating graphene as a thin film [10,13,30,39] have used the transfer matrix method, with each graphene layer represented by a individual matrix. This strategy is equivalent to the one we proposed using Equation (Equation 2), given that the multiplication of transfer matrices of the N-layers of graphene ultimately leads to an effective layer of thickness $L=N\times d_{g}=N\times 0.34$ nm, with the same refractive index $n_{3}^{g}$ of the monolayer. In the 2D model, we take the optical conductivity of multilayer graphene to be $\sigma_{t}=N\times \sigma_{g}$, where interlayer interactions are neglected (this has been shown experimentally to be a rather accurate description of multilayer graphene conductivity in the visible range [22]). Figure 4 shows the obtained ΔS as functions of the number of graphene layers. When graphene is modeled through the 2D or the 3D anisotropic models (green and cyan lines in Figure 4, respectively), the addition of graphene layers practically does not affect the sensitivity of the system based only on gold’s SPR. However, when the 3D isotropic model is applied, the increase is rather large (dark blue line, Figure 4). Additionally, by changing the graphene refractive index to the one used in [10,30] ($n_{3}^{g}=3+1.1491i$), we get an even larger sensitivity increase (red line, Figure 4). This can be perfectly explained if we bear in mind that the theory of SPPs in metals with thin dielectric coatings [40,41] predicts that the field of the SPPs at the metal/coating interface increases with the thickness of the dielectric film, if the refractive index of the coating is larger than that of the overlayer (in our case, the analyte), and decreases otherwise. Thus, the reported enhanced sensitivity for the thin isotropic graphene case looks rather trivial given that more intense fields, such as the case studied here ($n_{3}^{g}>n_{4}$), means augmented overlap integrals [42] and consequently increased sensitivity. This situation, however, requires an isotropic coating, which is not the case of graphene.

In essence, the problem of the isotropic model is that it assumes the electronic response of graphene to light to be independent of light’s electric field orientation. However, graphene’s out-of-plane conductivity significantly differs from the in-plane conductivity. As a matter of fact, since the early days of graphene isolation, it has been well known that the optical response of few-layer graphene in the visible range can be taken to be that of a stack of non-interacting 2D electron gas sheets [43]. For example, the opacity of graphene linearly increases with the number of layers [22], where each layer absorbs 2.3%. The optical contrast between graphene and substrates also increases linearly with the number of layers [23]. These results can be explained by the low value of the interlayer hopping energy (∼0.3 eV) that can be considered negligible when interlayer transitions are taken into account in the visible range. On the other hand, all forms of graphite are uniaxial with the c-axis perpendicular to the graphene plane [44]. Thus, there is no reason to treat graphite, few-layer graphene, or monolayer graphene as isotropic films.

## 4. Conclusions

In this paper, we theoretically studied the influence of graphene modeling to the sensitivity obtained in a refractometer based on gold’s SPR. When graphene is modeled as a surface (2D model) or an anisotropic thin film (anisotropic 3D model) the system’s sensitivity is practically unaffected by the presence and number of layers of graphene. However, when graphene is modeled as an isotropic thin film (isotropic 3D model), a significant non-physical sensitivity increase is observed. Therefore, we suggest that graphene modeling through the 2D or 3D anisotropic models are more accurate alternatives, which must be adopted for the study of graphene-assisted SPR sensors. It is important to highlight that by no means are we suggesting that graphene cannot effectively improve the sensitivity or the performance of SPR sensors by other means, in the visible range, as recently reported [45,46,47]. Our work is a call to look for the actual role played by graphene in the SPR sensors; for example, the enhanced sensitivity of SPR sensors with graphene oxide (GO) is attributed to the large surface area and molecule adsorbability of GO [48,49] without invoking its electromagnetic properties. Our point can also be extended to the use of other 2D materials in SPR sensors, where the sensitivity is also predicted to increase with the number of layers [19,20,21].

## Figures and Tables

**Figure 1 nanomaterials-12-02562-f001:**
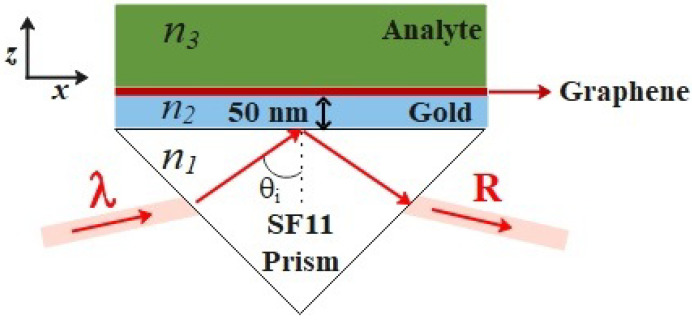
Schematic representation of the SPR sensor.

**Figure 2 nanomaterials-12-02562-f002:**
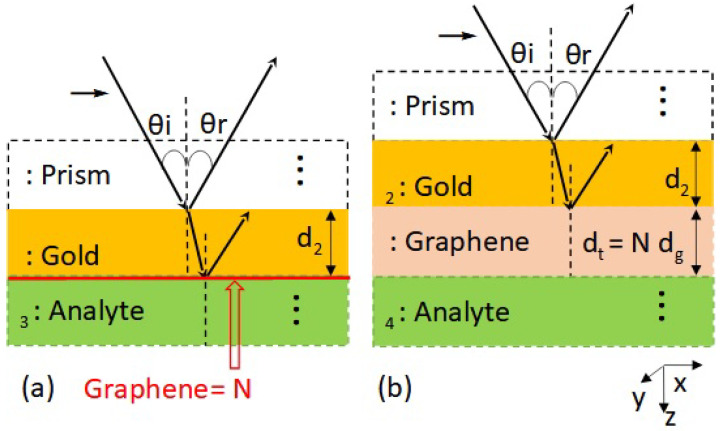
(**a**) Three-layer system: graphene as an atomic sheet with a surface conductivity $\sigma_{g}$. (**b**) Four-layer system: graphene as a thin layer film with thickness d${}_{g}$ and refractive index $n_{3}^{g}$.

**Figure 3 nanomaterials-12-02562-f003:**
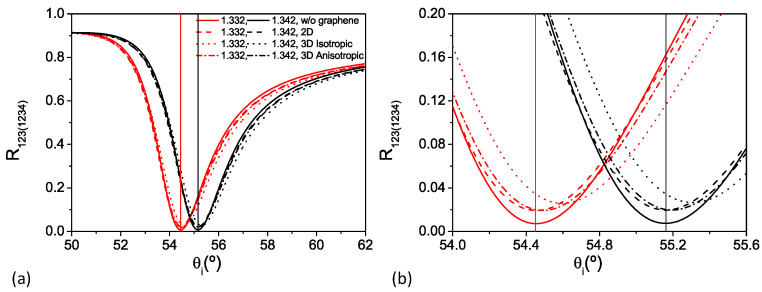
(**a**) Calculated reflectance $\left(R_{123(1234)}\right)$ as a function of the incident angle $\left(\theta_{i}\right)$ for analyte refractive indices of 1.332 and 1.342 (red and black curves, respectively). The vertical lines indicate the positions of $\left(\theta_{SPR}\right)$ for the system without graphene. Graphene modeled as a surface conductivity (dashed lines); and as isotropic $\left(n_{3}^{g}\right)$ (dotted lines) and anisotropic $\left(n_{3\mathrm{in}}^{g}\right)$ (dashed-dotted lines) films. (**b**) Zoom in the the minimum reflectance region of (**a**).

**Figure 4 nanomaterials-12-02562-f004:**
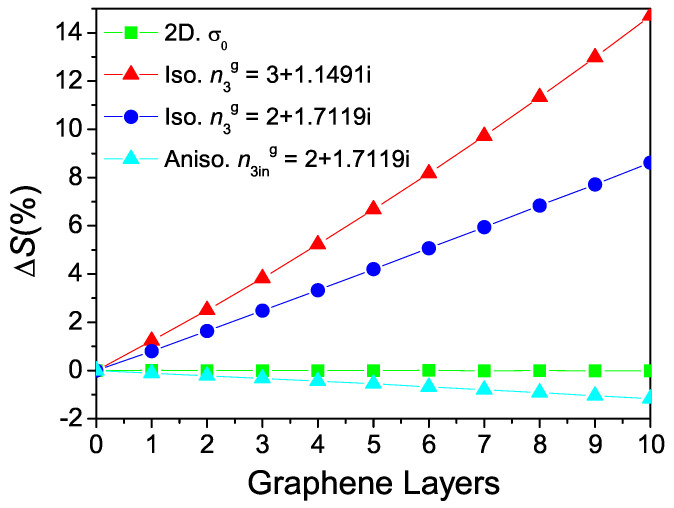
Calculated sensitivity increase for the addition of graphene layers compared with the system based only on gold. Graphene as an: atomic sheet (green), isotropic film, $n_{3}^{g}=3+1.1491i$ (dark blue) and $n_{3}^{g}=2+1.7119i$ (red); anisotropic film, $n_{3\mathrm{in}}^{g}=2+1.7119i$ (cyan).

**Table 1 nanomaterials-12-02562-t001:** Sensitivity and sensitivity increase for the SPR sensor without and with a monolayer of graphene modeled in the three different examined ways.

Model	$S_{\mathrm{Au}(\mathrm{gr})}{(}^{\circ }/\mathrm{RIU})$	ΔS (%)
Gold	70.8951	0
2D	70.8962	0.0016
3D Iso.	71.4515	0.7848
3D Aniso.	70.8339	−0.086

## Data Availability

Data is contained within the article.

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
