# Peer review of "A Critical Analysis on the Sensitivity Enhancement of Surface Plasmon Resonance Sensors with Graphene"

_nanomaterials, 2022, doi:10.3390/nano12152562_

Round 1

Reviewer 1 Report

This work reports modelling results  to analyze the role of graphene layer as surface and anisotropic thin film or isotropic one on SPR sensitivity enhancement. This paper seems quite interesting, and it is well organized and well described.

Anyway, I suggest extending the refractive index range in the calculation (up to 1.37) to support modelling results.

I also suggest to choose a refractive index (RI) of the prism lower than the one of SF11, i.e. BK7 , for modelling results. In this last case, a larger shift in the dip angle should be expected and the comparison among Sensitivity and sensitivity increase for the SPR sensor without and with a monolayer of graphene modelled in the three different cases could be probably enhanced.

Reviewer 2 Report

The authors have demonstrated the modeling of SPR by using 2D and 3D graphene. They tell the accurate modeling is better to be implemented by using 2D and 3D anisotropic model which is used to represent graphene. This manuscript could be accepted once the following issues have been addressed.

1.      What are the difference for modeling, such as structure, material,  between 2D model and 3D isotropic case shown in Figure 3?

2.      The author is suggested to tell how to define the refractive index of graphene and what are the value of parameter in Eq.(4).

3.      The language should be improved, such as in page 1, line 22, these waves are highly sensitive to changes in dielectric function the surrounding environment.

Page 2 line 45 to access the actual impact of graphene on the SPR sensor sensitivity we model graphene in a number of ways and compare the obtained results.

Reviewer 3 Report

The paper presents a critical review on surface plasmon sensors that uses graphene layer for enhancing performance. Mainly, the paper brings attention to the less-than-ideal ways that other researchers have modeled graphene. The presented data shows that the sensor performance only enhances when graphene is modeled as an isotropic layer. For more realistic models, it is shown that the performance significant is negligible. The paper is well written, clear, and covers an important topic. Indeed, current scientific literature is covered with application of graphene in many applications. Most of these works are simulation based. Due to the buzz around graphene, many subpar modeling papers are unfortunately published. I highly recommend this paper to be published as it is rare to see a critical paper on the matter.

A few minor comments:

1.       It might be good to include a brief discussion on why the isotropic model is not appropriate for graphene.

2.       Is it possible to make an optimized structure (while keeping an anistorpic model for graphene) that produces substantial improvement in sensing performance?

3.       Resolution of Fig. 3 should be improved.

Round 2

Reviewer 1 Report

it is ok for me